# The Role of Hydrogen Sulfide (H_2_S) in Epigenetic Regulation of Neurodegenerative Diseases: A Systematic Review

**DOI:** 10.3390/ijms241612555

**Published:** 2023-08-08

**Authors:** Bombonica Gabriela Dogaru, Constantin Munteanu

**Affiliations:** 1Faculty of Medicine, “Iuliu Hatieganu” University of Medicine and Pharmacy, 400012 Cluj-Napoca, Romania; dogarugabrielaumf@gmail.com; 2Clinical Rehabilitation Hospital, 400437 Cluj-Napoca, Romania; 3Teaching Emergency Hospital “Bagdasar-Arseni” (TEHBA), 041915 Bucharest, Romania; 4Faculty of Medical Bioengineering, University of Medicine and Pharmacy “Grigore T. Popa” Iași, 700454 Iași, Romania

**Keywords:** hydrogen sulfide, epigenetic regulation, neurodegenerative diseases, DNA methylation, histone modifications, non-coding RNAs, Alzheimer’s disease, Parkinson’s disease, Huntington’s disease

## Abstract

This review explores the emerging role of hydrogen sulfide (H_2_S) in modulating epigenetic mechanisms involved in neurodegenerative diseases. Accumulating evidence has begun to elucidate the multifaceted ways in which H_2_S influences the epigenetic landscape and, subsequently, the progression of various neurodegenerative disorders, including Alzheimer’s, Parkinson’s, and Huntington’s disease. H_2_S can modulate key components of the epigenetic machinery, such as DNA methylation, histone modifications, and non-coding RNAs, impacting gene expression and cellular functions relevant to neuronal survival, inflammation, and synaptic plasticity. We synthesize recent research that positions H_2_S as an essential player within this intricate network, with the potential to open new therapeutic avenues for these currently incurable conditions. Despite significant progress, there remains a considerable gap in our understanding of the precise molecular mechanisms and the potential therapeutic implications of modulating H_2_S levels or its downstream targets. We conclude by identifying future directions for research aimed at exploiting the therapeutic potential of H_2_S in neurodegenerative diseases.

## 1. Introduction

Neurodegenerative diseases, including Alzheimer’s, Parkinson’s, and Huntington’s, pose a formidable challenge to the modern medicine [1,2]. These conditions are characterized by the relentless and irreversible loss of neurons, leading to a gradual decline in cognitive and motor functions [3]. They have become a significant subset of non-communicable diseases, exacerbated by our longer human lifespan [4], impacting the lives of millions of individuals worldwide. Not only do these diseases cause emotional distress, but they also impose substantial economic burdens on society [5].

Delving into the biology of these complex conditions reveals an intricate web of causative factors involving a complex interplay of genetic, epigenetic, and environmental influences that collectively drive disease onset and progression [6,7,8,9]. Among these factors, epigenetic changes have emerged as critical determinants in the development and course of neurodegenerative diseases [10,11,12,13,14].

Epigenetic mechanisms play a fundamental role in gene regulation by facilitating dynamic changes in gene activity without altering the underlying DNA sequence [15,16,17,18,19,20,21,22,23,24]. Epigenetics includes DNA methylation, histone modification, and the non-coding RNAs [25,26,27,28,29,30].

Beyond their biological significance, epigenetic changes also offer an intriguing evolutionary perspective [31,32,33]. Modifying gene activity in response to environmental cues without altering the DNA sequence provides organisms with a remarkable evolutionary advantage [34,35,36,37]. It has likely allowed living organisms to adapt and thrive in diverse environments [38,39,40,41,42].

Despite its notorious association with the smell of rotten eggs and its potential for toxicity in high concentrations, H_2_S has emerged as a molecule of interest in studying many physiological and pathological processes [43,44,45,46,47,48,49,50,51]. H_2_S is endogenously produced in the brain through the enzymatic breakdown of cysteine by cystathionine β-synthase (CBS), cystathionine γ-lyase (CSE), and 3-mercaptopyruvate sulfurtransferase (3-MST) [52]. It acts as a neuromodulator and has been shown to play a pivotal role in regulating synaptic transmission, neuronal survival, and neuroinflammation [53,54].

Recent evidence suggests that H_2_S could interact with the various epigenetic mechanisms involved in neurodegenerative diseases [55,56,57,58,59,60]. By influencing epigenetic changes, H_2_S could impact gene expression patterns relevant to these diseases. This potential interaction between H_2_S and the epigenetic landscape provides a fresh perspective into our understanding of these complex conditions and highlights the need for further research into the role and mechanisms of H_2_S in these diseases [61,62,63,64,65]. Therefore, this review aims to provide a comprehensive overview of the current research exploring the role of H_2_S in regulating epigenetic processes associated with neurodegenerative diseases (Figure 1).

One of the key epigenetic mechanisms regulated by H_2_S is histone modification. H_2_S has been shown to modify histone proteins through sulfhydration, a process by which a sulfur atom is added to specific cysteine residues of histones [45]. Sulfhydration of histones can modulate chromatin structure and gene expression, ultimately influencing various cellular processes in neurons [66]. For instance, H_2_S-mediated sulfhydration of histones has been reported to affect the expression of genes involved in synaptic plasticity, memory formation, and neuronal survival. Dysregulation of this process has been implicated in the pathogenesis of neurodegenerative diseases (NDs) [67], including Alzheimer’s disease (AD) [68], Parkinson’s disease (PD), and Huntington’s disease (HD).

DNA methylation is another critical epigenetic mechanism influenced by H_2_S [69]. DNA methylation involves the addition of a methyl group to cytosine residues in CpG dinucleotides, leading to transcriptional repression of target genes. H_2_S has been shown to regulate the activity of DNA methyltransferases (DNMTs), the enzymes responsible for DNA methylation. Changes in DNMT activity due to H_2_S dysregulation have been associated with altered DNA methylation patterns in neurons, contributing to the aberrant gene expression observed in NDs. For instance, H_2_S-mediated changes in DNA methylation have been linked to the dysregulation of genes involved in neuroinflammation, oxidative stress, and neuronal survival [70,71,72,73].

In addition to histone modification and DNA methylation, H_2_S interacts with non-coding RNAs, including microRNAs (miRNAs) and long non-coding RNAs (lncRNAs) [74]. miRNAs are small non-coding RNAs that post-transcriptionally regulate gene expression by targeting mRNAs for degradation or translational repression. The dysregulation of miRNAs has been implicated in various aspects of neurodegeneration, including protein aggregation, neuroinflammation, and synaptic dysfunction. H_2_S has been shown to modulate the expression and activity of specific miRNAs, leading to altered gene expression profiles in neurons [75,76].

Furthermore, lncRNAs, a class of non-coding RNAs longer than 200 nucleotides, have also been found to play crucial roles in neurodegenerative processes. H_2_S can influence the expression and function of lncRNAs, thereby affecting gene expression and cellular processes in neurons. Dysregulation of specific lncRNAs has been associated with NDs, and their interaction with H_2_S further highlights the significance of epigenetic regulation in the neurodegeneration [77,78].

## 2. Methods

Our systematic review followed PRISMA guidelines and registered on PROSPERO ID: 449843. To ensure comprehensive coverage of the relevant literature, we searched multiple databases, including PubMed, Scopus/Elsevier, Web of Science, and Google Search. The search strategy involved the use of specific keywords and medical subject headings (MeSH) related to hydrogen sulfide, epigenetic regulation (DNA methylation, histone modifications, non-coding RNAs), and neurodegenerative diseases (such as Alzheimer’s, Parkinson’s, and Huntington’s diseases). After removing non-eligible and duplicate references, our review included 115 relevant studies. In addition to the databases mentioned above, we also searched https://clinicaltrials.gov/ (accessed on 1 June 2023) to identify potential clinical trials related to our topic (Figure 2).

## 3. Results and Discussion 

### 3.1. Hydrogen Sulfide and Neurodegenerative Diseases: An Overview

Hydrogen sulfide (H_2_S) is a multifaceted gasotransmitter, a gas molecule that occurs naturally in organisms and has been recognized for its diverse roles in physiological and pathological processes [79,80,81]. Despite its association with the unpleasant smell of rotten eggs, H_2_S has garnered increasing attention in biology [48,82]. H_2_S has been found to influence neuroinflammation, oxidative stress, and mitochondrial dysfunction, all of which are implicated in the pathogenesis of neurodegenerative disorders. Its significance in neurodegeneration is an intriguing area of investigation, with the potential to reveal new therapeutic strategies [54,83,84]. By contextualizing the study of H_2_S within the framework of evolution and epigenetics, we obtain a deeper understanding of its complex physio-pathology [45,85,86,87].

This scientific exploration highlights the importance of interdisciplinary research, bridging evolutionary biology and molecular medicine, and may pave the way for improved treatments and better quality of life for individuals impacted by neurodegenerative disorders and potentially other health conditions.

In the central nervous system (CNS), H_2_S regulates vasodilation, protects against oxidative stress-induced damage, and modulates inflammatory responses, vital for maintaining neuronal health and adequate brain function [88,89]. H_2_S also influences proper immune system functioning, as it acts as an anti-inflammatory agent, dampening excessive inflammation and promoting immune balance [90]. However, it is essential to tightly control H_2_S levels, as high concentrations can become toxic, leading to cellular damage and death [54,91]. Enzymes such as CBS, CSE, and 3-MST regulate H_2_S levels to prevent harmful accumulation while allowing for its beneficial signaling functions [84,92]. The study of H_2_S in biology is placed within the context of evolution, suggesting that its role in cellular processes has likely evolved to help organisms adapt to changing environments and cope with environmental challenges, thus influencing gene expression and cellular functions in various ways [32,93,94].

In Alzheimer’s disease (AD), H_2_S has been shown to modulate the activity of enzymes involved in amyloid-beta (Aβ) production and tau protein phosphorylation, critical processes in AD pathogenesis [54,95]. Specifically, H_2_S can promote the production of Aβ through its influence on the enzyme beta-secretase (BACE1) and the gamma-secretase complex [96,97]. Additionally, H_2_S has been shown to induce tau phosphorylation, leading to the aggregation of hyperphosphorylated tau into neurofibrillary tangles [98]. Furthermore, H_2_S can contribute to neuroinflammation and oxidative stress, exacerbating neurodegeneration in AD [99].

In Parkinson’s disease (PD), the aggregation of alpha-synuclein protein into Lewy bodies is a central feature of the disease [100]. H_2_S has been implicated in the assembly and misfolding of alpha-synuclein, promoting its neurotoxicity and contributing to the progression of PD. Moreover, H_2_S can affect mitochondrial function and induce oxidative stress, both associated with dopaminergic neuronal death in PD [101,102,103]. H_2_S-induced inflammation and microglial activation may also affect disease pathogenesis [83,103,104].

In Huntington’s disease (HD), accumulating mutant huntingtin protein with an expanded polyglutamine repeat is critical in the disease process [105]. H_2_S has been shown to influence the aggregation and toxicity of mutant huntingtin, contributing to the degeneration of neurons in the striatum and other brain regions affected in HD. Additionally, H_2_S can exacerbate mitochondrial dysfunction and oxidative stress, further contributing to neuronal damage in HD [106,107].

Emerging research also suggests that H_2_S may be involved in the dysregulation of autophagy, a cellular process crucial for removing misfolded proteins and damaged organelles. Dysfunctional autophagy has been implicated in the pathogenesis of neurodegenerative diseases, and H_2_S may contribute to autophagic impairments in these conditions [108,109,110,111]. Although the precise mechanisms by which H_2_S exerts its effects in neurodegenerative diseases are still under investigation, the emerging evidence highlights its potential as a promising therapeutic target.

### 3.2. Epigenetic Regulation in Neurodegenerative Diseases

Epigenetic regulation plays a pivotal role in the pathogenesis of neurodegenerative diseases, influencing gene expression and cellular functions relevant to neuronal health [112]. Epigenetics refers to modifications that occur on the genome without altering the underlying DNA sequence, and these changes can be inherited or influenced by environmental factors. In neurodegenerative diseases, dysregulation of epigenetic mechanisms has been implicated in disrupting normal cellular processes and the progressive loss of neurons [16,113].

Chromatin remodeling is a fundamental epigenetic mechanism that regulates gene expression by modifying the chromatin structure, comprising DNA and histone proteins [114]. ATP-dependent chromatin remodeling complexes can lead to the misregulation of crucial genes involved in neuronal survival and function [115]. Moreover, histone modifications, such as acetylation and methylation, dynamically regulate gene expression in neurons, and their perturbations have been observed in various neurodegenerative conditions [116]. These epigenetic alterations in chromatin remodeling can impact the expression of genes associated with disease pathology, highlighting the significance of chromatin remodeling in the neurodegeneration [117]. Targeting chromatin remodeling factors may hold promise for developing epigenetic-based therapies to counteract neurodegenerative disease progression and promote neuroprotection. Further research is needed to elucidate the precise molecular mechanisms and potential therapeutic implications of chromatin remodeling in neurodegenerative diseases [28].

Three primary epigenetic mechanisms are particularly relevant to neurodegeneration: DNA methylation; histone modification; and non-coding RNAs [25].

#### 3.2.1. DNA Methylation

DNA methylation involves adding a methyl group to specific cytosine residues in the DNA sequence, typically occurring at CpG sites (Cytosine-phosphate-Guanine) [118,119,120]. Methylation of CpG islands in the promoter regions of genes is associated with gene silencing, leading to reduced gene expression. In neurodegenerative diseases, aberrant DNA methylation patterns have been observed in genes that play crucial roles in neuronal function, such as synaptic plasticity, neuroinflammation, and oxidative stress response. These changes in DNA methylation can impact the expression of genes linked to disease pathogenesis, contributing to the dysfunction and death of neurons [120].

Emerging research has shed light on the dynamic nature of DNA methylation in neurodegenerative disorders and its impact on disease progression [121]. For instance, studies have shown altered DNA methylation patterns in genes associated with amyloid-beta processing and tau phosphorylation in Alzheimer’s disease (AD) [122]. Similarly, in Parkinson’s disease (PD) [123], dysregulated DNA methylation has been observed in genes linked to mitochondrial function, dopamine signaling, and neuroinflammation [124].

Moreover, DNA methylation changes have been implicated in regulating genes involved in response to oxidative stress, a process closely linked to neurodegeneration. Oxidative stress-induced DNA methylation alterations can affect the expression of antioxidant defense genes, exacerbating neuronal vulnerability to oxidative damage [125].

Advancements in epigenomic technologies, such as genome-wide DNA methylation profiling, have provided valuable insights into the specific genes and pathways affected by DNA methylation changes in neurodegenerative diseases [119].

#### 3.2.2. Histone Modification

Histones are proteins around which DNA is wrapped to form chromatin, the complex structure that packages DNA within the cell nucleus. Histone modifications, such as acetylation, methylation, phosphorylation, and ubiquitination, can alter the accessibility of DNA to transcriptional machinery, affecting gene expression. Dysregulation of histone modifications has been involved in neurodegenerative diseases, leading to altered gene expression patterns that might contribute to disease progression. For example, histone deacetylases (HDACs), enzymes involved in histone deacetylation, have been shown to regulate gene expression in Alzheimer’s and Huntington’s disease [126].

In Alzheimer’s disease (AD), perturbations in histone acetylation and deacetylation processes have been linked to disease pathology. Histone deacetylases (HDACs), a class of enzymes responsible for histone deacetylation, play a crucial role in regulating gene expression in AD [117]. Studies have shown that dysregulation of specific HDACs, such as HDAC2, is associated with synaptic dysfunction and cognitive impairment in AD. Additionally, histone acetyltransferases (HATs), the enzymes responsible for histone acetylation, have been implicated in AD pathogenesis. HATs are involved in the acetylation of histones, leading to a relaxed chromatin structure and increased gene transcription. Notably, dysregulation of HATs may contribute to the altered expression of genes involved in neuroinflammation and amyloid-beta processing [116].

Moreover, histone modifications have been linked to other neurodegenerative diseases, such as Parkinson’s disease (PD) [127]. In PD, altered histone acetylation levels have been associated with mitochondrial dysfunction and oxidative stress, contributing to dopaminergic neuronal degeneration [128]. Additionally, histone methylation patterns have been reported to regulate alpha-synuclein expression, a protein implicated in PD pathology [60].

In Huntington’s disease (HD), an inherited neurodegenerative disorder, histone modifications have also been implicated in disease pathophysiology. For instance, aberrant histone methylation patterns have been observed in HD, leading to changes in gene expression associated with neuronal dysfunction. Furthermore, HDAC inhibitors have shown promising effects in preclinical models of HD, indicating the therapeutic potential of targeting histone modifications in this disorder [129].

#### 3.2.3. Non-Coding RNAs

Non-coding RNAs (ncRNAs) are RNA molecules that do not code for proteins but have regulatory functions in the cell. Two major types of ncRNAs involved in epigenetic regulation are microRNAs (miRNAs) and long non-coding RNAs (lncRNAs).

MiRNAs are small RNA molecules that can bind to target messenger RNAs (mRNAs), leading to mRNA degradation or translational repression. The dysregulation of miRNAs has been linked to neurodegenerative diseases. Aberrant expression of specific miRNAs can disrupt key pathways related to neuroinflammation, synaptic plasticity, and mitochondrial function, contributing to the pathogenesis of neurodegenerative diseases.

LncRNAs, on the other hand, are a diverse group of transcripts that are longer than 200 nucleotides and do not encode proteins [130]. LncRNAs can interact with chromatin-modifying complexes, influencing gene expression by epigenetic mechanisms. Altered expression of lncRNAs has been associated with neurodegenerative disorders, contributing to the dysregulation of gene expression and cellular functions [131,132,133,134].

Accumulating evidence has revealed the pivotal roles of lncRNAs in epigenetic regulation, acting as scaffolds for chromatin-modifying complexes or interacting with various epigenetic regulators to influence gene expression [135].

In neurodegenerative diseases, altered expression of lncRNAs has been associated with dysregulated gene expression patterns and cellular dysfunctions [136]. For instance, some lncRNAs have been found to interact with histone-modifying enzymes, such as histone methyltransferases or demethylases, leading to changes in histone methylation patterns and subsequent transcriptional alterations [25].

Additionally, lncRNAs can function as competing endogenous RNAs (ceRNAs) by competitively binding to miRNAs, thereby modulating the availability of miRNAs for target mRNAs. This ceRNA crosstalk may play a critical role in fine-tuning gene expression networks in the context of neurodegeneration [137].

Dysregulated lncRNA-miRNA interactions have been reported in neurodegenerative diseases, and their effects on target gene expression may impact pathways involved in neuronal survival, neuroinflammation, and protein aggregation [138].

Moreover, recent studies have highlighted the involvement of circular RNAs (circRNAs) in neurodegenerative diseases. CircRNAs are a unique class of ncRNAs with covalently closed circular structures. They have been shown to regulate gene expression by interacting with miRNAs or RNA-binding proteins, and their dysregulation has been implicated in the pathogenesis of neurodegenerative disorders [139].

The dynamic nature of epigenetic modifications presents opportunities for therapeutic interventions, as these changes are potentially reversible. Targeting epigenetic mechanisms holds promise for developing novel therapies to modify disease progression and improve the outcomes for individuals affected by neurodegenerative diseases. However, a comprehensive understanding of the specific epigenetic changes and their functional consequences in different neurodegenerative disorders remains an area of active research. Unraveling the complexities of epigenetic regulation in these diseases may lead to identifying biomarkers and novel therapeutic targets, ultimately providing hope for more effective treatments in the future [121,140,141].

Given the complex nature of neurodegenerative diseases, including Alzheimer’s, Parkinson’s, and Huntington’s, it is evident that epigenetic regulation plays a significant role in disease progression. The intricate interplay between genetic, environmental, and epigenetic factors contributes to the loss of neurons observed in these disorders.

H_2_S has been found to interact with various epigenetic mechanisms, influencing gene expression and cellular functions relevant to neuronal health. By interacting with key epigenetic regulators such as DNA methylation, histone modifications, and non-coding RNAs, H_2_S can influence the expression of genes crucial for neuronal function and survival [8,142,143].

### 3.3. Hydrogen Sulfide and DNA Methylation

The modulation of DNA methylation by hydrogen sulfide (H_2_S) represents a fascinating interplay between this gasotransmitter and epigenetic regulation in the context of neurodegenerative diseases. Studies have revealed that H_2_S can modulate DNA methylation patterns by affecting the activity of enzymes involved in DNA methylation, such as DNA methyltransferases (DNMTs). For instance, H_2_S has been shown to inhibit DNMT activity, resulting in decreased DNA methylation at specific gene promoter regions. This reduced methylation can lead to altered gene expression, potentially impacting pathways crucial to neuronal survival, neuroinflammation, and oxidative stress response. Moreover, H_2_S has been found to influence the expression of genes’ expression in regulating H_2_S metabolism, creating a feedback loop that further impacts the epigenetic landscape. This intricate interplay between H_2_S and DNA methylation highlights the potential importance of epigenetic mechanisms in the pathogenesis of neurodegenerative disorders, offering new avenues for therapeutic interventions targeting H_2_S-mediated epigenetic dysregulation. Further research in this area may unveil the full extent of H_2_S’s role in shaping the epigenetic landscape and its implications for neurodegenerative disease progression and potential treatment strategies [51,144,145,146,147,148] (Table 1).

### 3.4. Hydrogen Sulfide and Histone Modifications

H_2_S exerts its effects on histones through interactions with histone-modifying enzymes, affecting histones’ acetylation, methylation, and phosphorylation. By influencing these histone modifications, H_2_S can modulate the accessibility of DNA to transcriptional machinery, leading to changes in gene expression patterns. Notably, H_2_S has been shown to impact the activity of histone acetyltransferases (HATs) and histone deacetylases (HDACs), enzymes involved in histone acetylation, which play crucial roles in regulating gene expression in neurodegenerative diseases. Dysregulation of histone modifications by H_2_S may contribute to altered expression of genes linked to neuroinflammation, neuroprotection, and other processes involved in neurodegeneration. Further investigations into the specific molecular interactions between H_2_S and histone-modifying enzymes will be crucial for unraveling the complex mechanisms underlying H_2_S-mediated epigenetic regulation in the context of neurodegeneration, potentially leading to the development of novel epigenetic-based interventions for neurodegenerative diseases [45,84,149,150,151,152] (Table 2).

### 3.5. Hydrogen Sulfide and Non-Coding RNAs

H_2_S has been shown to modulate the expression of specific microRNAs (miRNAs) and long non-coding RNAs (lncRNAs) that play regulatory roles in gene expression. By influencing the levels of these ncRNAs, H_2_S can affect the stability of mRNAs and protein translation, leading to changes in cellular functions. Dysregulation of miRNAs and lncRNAs has been observed in neurodegenerative diseases, and the interplay between H_2_S and these ncRNAs may contribute to disease pathogenesis [75,146,153,154] (Table 3).

### 3.6. Future Directions

Future research efforts will be pivotal in advancing the potential of H_2_S-based therapies for neurodegenerative diseases, offering new hope to patients facing these devastating disorders. To achieve this objective, several crucial areas require investigation. Firstly, elucidating the precise molecular mechanisms through which H_2_S interacts with epigenetic regulation and cellular pathways in neurodegeneration is essential to fully understanding its neuroprotective effects and therapeutic applications. Secondly, comprehensive studies assessing long-term safety and efficacy are necessary before translating H_2_S-based therapies to clinical settings. Understanding potential side effects, dose-response relationships, and effects on cellular processes will ensure the therapies’ safety and effectiveness. Thirdly, it is crucial to identify optimal delivery methods for H_2_S-based therapies, considering their bioavailability and tissue distribution in different administration routes for varying disease stages and patient populations. Furthermore, targeted research is needed to determine the suitability of H_2_S-based therapies for specific neurodegenerative diseases, such as Alzheimer’s, Parkinson’s, and Huntington’s, and personalized medicine approaches should be explored to develop tailored therapies based on individual disease profiles and patient characteristics.

Investigating the potential synergistic effects of H_2_S-based therapies with existing treatments or emerging therapeutic agents may lead to innovative combination therapies that enhance neuroprotection and disease modification. Moving H_2_S-based therapies from preclinical research to clinical trials will require well-designed translational studies to establish safety, efficacy, and dosage recommendations. Neuroimaging techniques can provide valuable insights into the mechanisms of action and potential benefits of H_2_S-based therapies. Determining the optimal therapeutic window and identifying reliable biomarkers for monitoring treatment response are also crucial steps in advancing the field of H_2_S-based therapies for neurodegenerative diseases. Emphasizing research in these areas will pave the way for innovative and targeted therapies, bringing us closer to effective treatments for these debilitating conditions.

## 4. Conclusions

Epigenetic regulation has emerged as a critical determinant in the pathogenesis and progression of neurodegenerative diseases. The interaction between H_2_S and different epigenetic mechanisms, such as DNA methylation, histone modifications, and non-coding RNAs, suggests that H_2_S could influence gene expression and cellular functions relevant to neurodegenerative diseases. Understanding the precise molecular mechanisms underlying H_2_S’s interactions with epigenetic processes is essential to develop targeted and effective therapeutic strategies. Furthermore, investigating the long-term safety and efficacy of H_2_S-based therapies will be critical for their clinical translation.

Identifying optimal delivery methods, targeting disease-specific effects, and developing personalized medicine approaches will ensure the efficacy of H_2_S-based therapies for individual patients. Additionally, research on combination therapies and the development of reliable biomarkers for monitoring treatment response will further enhance the potential benefits of H_2_S interventions.

Overall, exploring H_2_S’s role in epigenetic regulation and neurodegeneration represents a promising avenue for future research. Advancements in this field have the potential to revolutionize the treatment landscape for neurodegenerative diseases, offering new hope to patients and their families facing these currently incurable conditions.

## Figures and Tables

**Figure 1 ijms-24-12555-f001:**
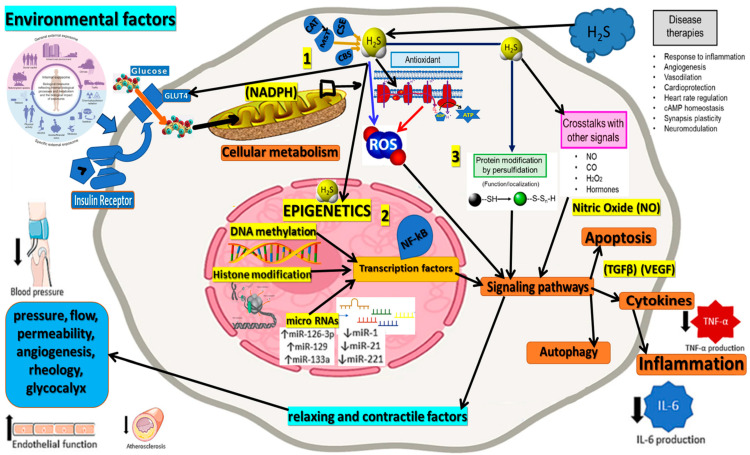
This figure illustrates the central role of Hydrogen Sulfide (H_2_S) in epigenetic regulation and its interactions with various processes associated with neurodegenerative diseases. H_2_S affects epigenetic mechanisms, including histone modification, DNA methylation, and non-coding RNAs, which modulate gene expression and cellular functions relevant to neurodegeneration. H_2_S influences reactive oxygen species (ROS) production and oxidative stress levels, which play a critical role in the pathogenesis of neurodegenerative disorders. By reducing ROS production (blue arrow) and inhibiting ROS-generating processes at the mitochondrial level (red arrows), H_2_S impacts a broad spectrum of biological functions, as depicted in this figure for illustrative purposes. The interplay between H_2_S, epigenetic processes, and oxidative stress offers valuable insights into the molecular mechanisms underlying neurodegeneration and highlights the potential for therapeutic interventions to restore epigenetic balance and mitigate oxidative stress to combat neurodegenerative diseases effectively. The figure corroborates 3 major parts: 1. external sources and internal production of H_2_S; 2. the epigenetic role of H_2_S; and 3. the physiological connections presented in a simplified way.

**Figure 2 ijms-24-12555-f002:**
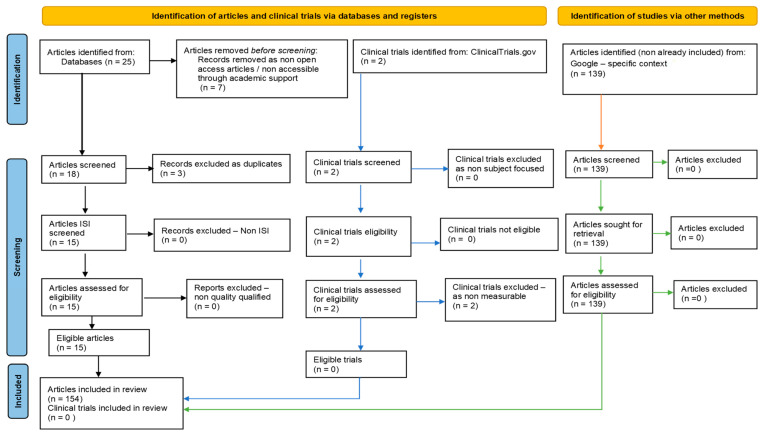
Adapted PRISMA flow diagram, customized for our study.

**Table 1 ijms-24-12555-t001:** Data regarding Hydrogen Sulfide and DNA Methylation in Neurodegenerative diseases.

Ref.	Title	Pathology	Extracted Data Regarding Epigenetics	Major Outcome
[51]	Therapeutic importance of hydrogen sulfide in age-associated neurodegenerative diseases	Neurodegenerative diseases	H_2_S, a crucial signaling molecule, regulates DNA methylation, an essential epigenetic modification impacting gene expression and cellular function. H_2_S influences DNA methylation in oxidative stress and aging conditions, safeguarding against DNA damage and preserving genomic integrity.	H_2_S modulates DNA methyltransferases.
[144]	Abnormal Homocysteine Metabolism: An Insight of Alzheimer’s Disease from DNA Methylation	Alzheimer’s disease	DNA methylation involves adding methyl groups to cytosine-phosphate-guanine (CpG) sequences catalyzed by DNMT enzymes. DNMT1 maintains existing methylation during cell division, while DNMT3a and DNMT3b create new methylation patterns on unmethylated DNA strands. H_2_S interference with these processes opens the potential for novel therapeutic strategies against neurodegenerative diseases.	In Alzheimer’s disease (AD), changes in DNA methylation impact the production of amyloid-beta (Aβ) plaques and tau hyperphosphorylation, key factors in AD pathology.
[145]	Hydrogen sulfide signalling in the CNS—Comparison with NO	Schizophrenia	In C3H mice, DNA methylation levels at the MPST gene were significantly increased and positively correlated with MPST expression. In schizophrenia patients, MPST levels were positively associated with symptom severity scores. In MPST-transgenic mice, genes related to energy formation were downregulated, and mitochondrial energy metabolism was impaired.	H_2_S involvement in DNA methylation regulation of the MPST gene in schizophrenia may shed light on the molecular basis of energy metabolism dysregulation in the CNS.
[146]	Hydrogen Sulfide Improves Angiogenesis by Regulating the Transcription of pri-miR-126 in Diabetic Endothelial Cells	Parkinson’s disease	DNA methylation, an essential epigenetic modification, regulates gene expression. In diabetic mice, DNMT1 overexpression reduces miR-126-3p levels, impairing endothelial cell function and blood flow recovery. Exogenous H_2_S reverses these effects by downregulating DNMT1 expression, enhancing miR-126-3p levels, and promoting angiogenesis.	The interplay between H_2_S and DNA methylation in the regulation of miR-126-3p expression.
[147]	Hydrogen sulfide in ageing, longevity, and disease	Alzheimer’s disease Parkinson’s disease	Metformin interacts with H_2_S signaling and activates AMPK, inhibiting mTOR and IIS signaling pathways. It removes homocysteine-stimulated hypermethylation of the CSE promotor region, resulting in increased CSE expression and H_2_S production.	Metformin’s ability to increase H_2_S levels is linked to its role in remodeling DNA methylation patterns.
[148]	Cell Rearrangement and Oxidant/Antioxidant Imbalance in Huntington’s Disease	Huntington’s disease	In HD, the toxic protein mHtt can interfere with the transcriptional machinery, altering histone modifications and DNA methylation and impairing gene expression and neuronal dysfunction.	HD is linked to accelerated epigenetic aging. Epigenetic clocks show a correlation between HD progression and epigenetic age.

**Table 2 ijms-24-12555-t002:** Data regarding Hydrogen Sulfide and Histone Modifications in Neurodegenerative Diseases.

Ref.	Title	Pathology	Extracted Data Regarding Epigenetics	Major Outcome
[45]	Hydrogen sulfide-induced post-translational modification as a potential drug target	Neurodegenerative disease, Alzheimer’s, Parkinson’s, Huntington’s diseases	H_2_S affects histones. S-sulfhydration of histones can regulate gene expression and epigenetic modifications. H_2_S-mediated S-sulfhydration of histone modifiers, such as Sirt1, affects aging, metabolism, and oxidative stress tolerance.	H_2_S-S-sulfhydration histone modifiers, such as Sirt1, influences aging, metabolism, and oxidative stress.
[84]	Exploring mitochondrial hydrogen sulfide signalling for therapeutic interventions in vascular diseases	Neurodegenerative diseases, Parkinson’s disease	H_2_S administration has been found to increase antioxidant proteins such as Trx-1 through the Nrf-2 pathway, leading to cardioprotection in ischemia-induced heart failure. H_2_S also regulates members of the SIRT family, such as SIRT1, SIRT3, and SIRT6, which play critical roles in histone and non-histone protein modifications. These findings highlight the importance of H_2_S-SIRT interactions in mediating cellular protection and physiological effects.	H_2_S-SIRT interactions mediate cellular protection, impacting histone modifications and cellular functions.
[149]	Hydrogen Sulfide Biology and Its Role in Cancer	Neurodegenerative diseases	H_2_S can increase E-cadherin levels, inhibit histone deacetylase, and modulate NF-κB signaling, resulting in anti-metastatic and tumor-suppressive effects. However, the exact molecular targets underlying H_2_S’s diverse effects on biological processes, including cancer, require further investigation. Chronic exposure to H_2_S or its derivatives may have detrimental effects, including NF-κB inhibition and apoptosis.	H_2_S influences histone deacetylase and acetyltransferase activities, impacting gene expression and chromatin structure.
[150]	Protective effect of hydrogen sulfide is mediated by negative regulation of epigenetic histone acetylation in Parkinson’s disease	Neurodegenerative diseases Parkinson’s disease	Histone modifications and DNA methylation have been linked to the pathogenicity of Parkinson’s disease (PD). Histone deacetylase (HDAC) enzymes mediate chromatin condensation and inhibit gene transcription, while histone acetyltransferases (HAT) reverse these effects. Imbalances in HDAC and HAT activities are associated with neurodegenerative diseases, including PD. Inhibiting HDAC has shown promise in rescuing cells from degeneration in PD models. This study investigated the impact of HDAC inhibitor TSA on 6-hydroxydopamine-induced neurotoxicity in PD animal models.	H_2_S negatively regulates histone acetylation, impacting gene expression and neuronal survival in PD.
[151]	One-carbon epigenetics and redox biology of neurodegeneration	Alzheimer’s disease Parkinson’s disease Amyotrophic lateral sclerosis	Histone proteins form the histone octamer around which DNA is wrapped to create nucleosomes. Post-translational modifications (PTMs) of histone tails, including acetylation and methylation, regulate chromatin structure and gene expression. Histone acetyltransferases (HATs) add acetyl groups to lysine residues, promoting transcription, while histone deacetylases (HDACs) remove acetyl groups, leading to chromatin compaction and transcriptional inhibition.	H_2_S modulates histone acetyltransferases (HATs) and histone deacetylases (HDACs), influencing gene expression and chromatin remodeling.
[152]	Brain energy rescue: an emerging therapeutic concept for neurodegenerative disorders of ageing	Neurodegenerative disorders	Histones, as crucial chromatin components, are subject to post-translational modifications, including acetylation, which influences gene expression. Hydrogen sulfide (H_2_S) plays a role in cellular energetics by regulating the availability of acetyl-CoA, a precursor of acetyl groups used in histone acetylation. H_2_S-related pathways, such as sirtuin 1 activation, mitochondrial function, and gut microbiota-produced short-chain fatty acids, also impact histone modifications, linking cellular energetics to epigenetic regulation. Targeting these mechanisms may hold therapeutic potential for neurodegenerative disorders.	H_2_S-related pathways influence histone modifications, linking cellular energetics to epigenetic regulation in neurodegenerative disorders.

**Table 3 ijms-24-12555-t003:** Data regarding Hydrogen Sulfide and Non-Coding RNAs in Neurodegenerative diseases.

Ref.	Title	Pathology	Extracted Data Regarding Epigenetics	Major Outcome
[146]	Hydrogen Sulfide Improves Angiogenesis by Regulating the Transcription of pri-miR-126 in Diabetic Endothelial Cells	Parkinson’s disease	MicroRNAs (miRNAs) are non-coding RNAs that modulate various cellular processes, including angiogenesis. Specific miRNAs, such as miR-126-3p, regulate angiogenesis in vascular endothelial cells. H_2_S is involved in miRNA transcription regulation, and the interplay between H_2_S and miRNAs is critical in cardiovascular disease pathophysiology. H_2_S has been shown to decrease cardiomyocyte apoptosis and impact Parkinson’s disease through miRNA regulation.	miRNAs such as miR-126-3p regulate angiogenesis—connected to PD
[75]	Regulating of LncRNA2264/ miR-20b-5p/IL17RD axis on hydrogen sulfide exposure-induced inflammation in broiler thymus by activating MYD88/NF-κB pathway	Neurodegenerative disorders	lncRNA2264/miR-20b-5p/IL17RD axis was identified as part of the H_2_S-induced thymic inflammatory response. NcRNAs, including miRNAs and lncRNAs, can be potential biomarkers of environmental chemical exposure. In this study, lncRNA-sequencing revealed differentially expressed lncRNAs and miRNAs in the H_2_S-exposed group compared to the control group. Notably, lncRNA2264 showed significant downregulation, and it was identified as a molecular sponge for miR-20b-5p. MiR-20b-5p, which plays a role in immune cell function and inflammation, was significantly increased after H_2_S exposure.	NcRNAs, including miR-20b-5p and lncRNA2264, were identified as part of the H_2_S-induced thymic inflammatory response.
[153]	Overview on hydrogen sulfide-mediated suppression of vascular calcification and hemoglobin/heme-mediated vascular damage in atherosclerosis	Neurodegenerative disorders	Epigenetic alterations, including DNA methylation and microRNAs (miRNAs), are implicated in atherosclerosis development and are linked to H_2_S pathways. H_2_S influences histone modifications, enhancing SIRT1 activity to reduce endothelial inflammation and foam cell formation, potentially reducing atherosclerotic plaque development. Targeting these epigenetic regulatory checkpoints holds promise for atherosclerosis therapy.	H_2_S-mediated epigenetic changes may alleviate atherosclerosis by modulating SIRT1 activity.
[154]	The emerging role of long non-coding RNAs and microRNAs in neurodegenerative diseases: A perspective of machine learning	Neurodegenerative disease Alzheimer’s Parkinson’s Huntington’s diseases	Neurodegenerative diseases (NDs) exhibit similar early symptoms, making their timely detection and differentiation crucial. Dysregulation of microRNAs and long non-coding RNAs is associated with NDs, highlighting their potential as diagnostic and therapeutic targets. Machine learning can effectively classify non-coding RNA expression profiles between healthy and affected individuals, aiding in accurate ND diagnosis with accuracy rates of 85% to 95%. Artificial intelligence offers a promising approach to enhance clinical diagnosis and early disease identification based on non-coding RNAs.	ncRNAs, potential diagnostic and therapeutic targets in neurodegenerative diseases, and machine learning improve ND diagnosis based on ncRNA expression.

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
