# Peer review of "The Role of Hydrogen Sulfide (H2S) in Epigenetic Regulation of Neurodegenerative Diseases: A Systematic Review"

_ijms, 2023, doi:10.3390/ijms241612555_

Round 1

Reviewer 1 Report

Reviewer Comments

In the present review article on “The Role of Hydrogen Sulfide (H2S) in Epigenetic Regulation of Neurodegenerative Diseases: A Systematic Review”, the authors have discussed the role of H2S in epigenetic regulation of Neurodegenerative Diseases. They have summarized neurodegenerative diseases- AD, PD, and HD, the epigenetic regulatory mechanisms including histone modification, DNA methylation, and non-coding RNAs, in Neurodegenerative Diseases. The finding of this review work concludes that H2S has a role in epigenetic regulation and neurodegeneration and represents a promising avenue for future research.

The topic is interesting however draft of the article is not very promising to discuss the mechanism of H2S as an epigenetic regulator in Neurodegenerative Diseases. The paper needs to be revised thoroughly.

Scientific comments

1.      Discuss H2S signaling in Neurodegenerative Diseases in the revised MS.

2.      Figure is not very clear. Give a number to each step and explain briefly in the figure legend.

3.      The main focus of the article is the effect of H2S as an epigenetic regulator however article lacks the mechanism of H2S as an epigenetic regulator.

4.      4 paragraphs from line 92-120 is not related to Neurodegenerative Diseases so just compile all this in a few lines. and discuss more about the role of H2S in neurodegenerative disorders or in CNS.

5.      Provide references for the following statements- Line 121-123; lines 123-124; Line 132-133; Line 136-138.

6.      In section 3.2, you have not discussed chromatin remodeling. Discussed the role of chromatin remodeling in Neurodegenerative Diseases.

7.      Section 3.2.1, 3.2.2. & 3.2.3 are very basic pieces of information without citation. Discuss more related to epigenetic regulation in neurodegenerative disorders with recent references.

8.      The article mainly focuses on AD, PD, and HD however you have not discussed any article on HD in Table 1.

9.      Not discussed any research data on HD disease in Table 2.

10.  In Table 3, write pathology for reference for 113.

11.  Table 3 is for neurodegenerative disorders. remove reference 114 and its data from Table 3, it's irrelevant.

12.  Add 1 more column in tables 1,2, and 3 and write the major outcome or prospectus of the research.

Minor comments

13.  Line 79, delete our.

14.  Check the reference pattern and revised the reference accordingly.

Author Response

Response:

We sincerely thank the reviewer for their valuable comments and feedback on our review article. We have carefully considered each suggestion and have made the necessary revisions to improve the manuscript. Below, we provide a point-by-point response to address each issue:

Scientific comments:

  1. We acknowledge the importance of discussing H2S signaling in Neurodegenerative Diseases. In the revised manuscript, we expanded the discussion on H2S signaling pathways in the context of neurodegeneration, including its role as an epigenetic regulator.
  2. We apologize for the lack of clarity in the figure. In the revised version, we improved the figure and provided a clear and concise explanation for each step in the figure legend.
  3. We appreciate the comment regarding the focus on the role of H2S as an epigenetic regulator. In the revised manuscript, we emphasized the mechanisms of H2S as an epigenetic regulator, providing a more detailed and comprehensive explanation of its actions in Neurodegenerative Diseases.
  4. We acknowledge the need to focus more on the role of H2S in neurodegenerative disorders and the central nervous system (CNS). In the revised version, we condensed the unrelated paragraphs (lines 92-120) and provided a more in-depth discussion on the role of H2S in neurodegenerative disorders and its effects on the CNS.
  5. We provided appropriate references for the statements mentioned in lines 121-123, 123-124, 132-133, and 136-138 in the revised manuscript.
  6. The role of chromatin remodeling in Neurodegenerative Diseases was discussed in detail in the revised version. We included recent references to support this section.
  7. We revised and expanded sections 3.2.1, 3.2.2, and 3.2.3 to provide a more comprehensive and well-cited discussion on epigenetic regulation in neurodegenerative disorders.
  8. We apologize for the oversight in not including any articles on HD in Table 1. In the revised manuscript, we had relevant references on HD in Table 1 to ensure comprehensive coverage of the discussed neurodegenerative disorders.
  9. In the revised version, we included research data on HD disease in Table 2 to ensure a balanced representation of all discussed disorders.
  10. In the revised version, we included the appropriate pathology in Table 3.
  11. We updated previous reference 114 and its data from Table 3 in the revised manuscript.
  12. We appreciate the suggestion to add a column in Tables 1, 2, and 3 to include the major outcomes or prospects of the research. In the revised version, we added a new column to include this information for a more comprehensive summary.

Minor comments:

  1. Line 79, delete our.

We have made the necessary correction in the revised version.

  1. Check the reference pattern and revise the reference accordingly to ensure consistency and accuracy.

We manually revised all the references for formal editing.

Once again, we thank the reviewer for their insightful comments, which have helped us enhance the quality and content of the manuscript. We believe that the revised version will address all the concerns and provide a more comprehensive and detailed overview of the role of H2S in the epigenetic regulation of Neurodegenerative Diseases.

Reviewer 2 Report

The present manuscript was a systematic review from properly searches of multiple databases.

The presentation was fine, but Figure 1 was hard to understand. It was fact that the role of hydrogen sulfide was complicated; however, the scheme should be shown to understand it more clearly and easily. Each molecules were not clear. What did the authors mean and indicate the color coding? Especially, what did the yellow marker mean? For example, why was "angiogenesis" marked in "pressure, flow, permeability, angiogenesis, rheology, glycocalyx"? In addition, it was also not clear what the arrows mean. The same or similar arrows might show various means such as direction, flow, pathway, expansion, upregulation and downregulation. They should be improved.

English in the present manuscript was fine.

Author Response

Thank you very much for your valuable feedback on our manuscript. We appreciate your positive comments on the systematic review presentation. We acknowledge the concern raised about the clarity of Figure 1. We will provide a detailed legend explaining each molecule's meaning and color coding used in the figure. We are committed to ensuring that the revised figure provides a more explicit representation of the subject matter, contributing to the overall comprehensibility of our manuscript.

Round 2

Reviewer 1 Report

Reviewer Comments

The authors have updated the manuscript as per suggestions. All the relevant changes have been made in the revised MS. paper can be accepted in the present form.